# Mobocertinib (TAK-788) in *EGFR* Exon 20 Insertion+ Metastatic NSCLC: Patient-Reported Outcomes from EXCLAIM Extension Cohort

**DOI:** 10.3390/jcm12010112

**Published:** 2022-12-23

**Authors:** Maria Rosario Garcia Campelo, Caicun Zhou, Suresh S. Ramalingam, Huamao M. Lin, Tae Min Kim, Gregory J. Riely, Tarek Mekhail, Danny Nguyen, Erin Goodman, Minal Mehta, Sanjay Popat, Pasi A. Jänne

**Affiliations:** 1Department of Medical Oncology, University Hospital Complex A Coruña, 15006 A Coruña, Spain; 2Department of Oncology, Shanghai Pulmonary Hospital, Shanghai 200433, China; 3Winship Cancer Institute, Emory University, Atlanta, GA 30322, USA; 4Takeda Development Center Americas, Inc., Lexington, MA 02421, USA; 5Department of Internal Medicine, Seoul National University Hospital, Seoul 03080, Republic of Korea; 6Department of Medicine, Memorial Sloan Kettering Cancer Center, New York, NY 10065, USA; 7Advent Health Orlando, Orlando, FL 32803, USA; 8Pacific Shores Medical Group, Long Beach, CA 90813, USA; 9Lung Unit, The Royal Marsden Hospital, London SW3 6JJ, UK; 10The Institute of Cancer Research, University of London, London SM2 5NG, UK; 11Department of Medical Oncology, Dana-Farber Cancer Institute, Boston, MA 02215, USA; 12Harvard Medical School, Boston, MA 02215, USA

**Keywords:** carcinoma, non-small cell lung, lung neoplasms, mutation, neoplasm metastasis, quality of life

## Abstract

Mobocertinib, an oral, first-in-class epidermal growth factor receptor (EGFR) tyrosine kinase inhibitor selective for *EGFR* exon 20 insertions (ex20ins), achieved durable responses in adults with previously treated *EGFR* ex20ins+ metastatic non-small cell lung cancer (mNSCLC) in the EXCLAIM extension cohort of a phase 1/2 study (N = 96; NCT02716116). We assessed patient-reported outcomes (PROs) with mobocertinib 160 mg once daily (28-day cycles) in EXCLAIM (N = 90) with the European Organisation for Research and Treatment of Cancer Core Quality-of-Life Questionnaire (EORTC QLQ-C30) v3.0, lung cancer module (QLQ-LC13), EuroQol-5 Dimensions-5 Levels (EQ-5D-5L) questionnaire, and selected PRO Version of the Common Terminology Criteria for Adverse Events (PRO-CTCAE) questionnaire. Median treatment duration was 6.8 (range, 0.0–18.8) months (median follow-up: 13.0 [0.7–18.8] months; data cutoff: 1 November 2020). Clinically meaningful improvements in lung cancer symptoms measured by EORTC QLQ-LC13 were observed for dyspnea (54.4% of patients), cough (46.7%), and chest pain (38.9%), evident at cycle 2 and throughout treatment (least-squares mean [LSM] changes from baseline: dyspnea, −3.2 [*p* = 0.019]; cough, −9.3 [*p* < 0.001]; chest pain, −8.2 [*p* < 0.001]). EORTC QLQ-C30 results indicated no statistically significant changes in global health status/quality of life (LSM change from baseline: −1.8 [*p* = 0.235]). On symptom scores, significant worsening from baseline was observed for diarrhea (LSM change from baseline: +34.1; *p* < 0.001) and appetite loss (+6.6; *p* = 0.004), while improvements were observed for dyspnea (LSM change from baseline: −5.1 [*p* = 0.002]), insomnia (−6.5 [*p* = 0.001]), and constipation (−5.7 [*p <* 0.001]). EQ-5D-5L health status was maintained. Common PRO-CTCAE symptoms were diarrhea, dry skin, rash, and decreased appetite (mostly low grade); in the first 24 weeks of treatment, 64.4% of patients had worsening diarrhea frequency and 67.8% had worsening dry skin severity. Overall, PROs with mobocertinib showed clinically meaningful improvement in lung cancer–related symptoms, with health-related quality of life maintained despite changes in some adverse event symptom scales.

## 1. Introduction

In addition to prolonging survival, preserving or improving health-related quality of life (HRQoL) and having the ability to perform normal daily activities are important for patients with non-small cell lung cancer (NSCLC) [1]. The availability of effective, tolerable treatments is critical to achieving these goals. Unlike NSCLC tumors with epidermal growth factor receptor gene (*EGFR*)-sensitizing mutations (exon 19 and exon 21), which respond to treatment with the approved tyrosine kinase inhibitors (TKIs) afatinib, erlotinib, and gefitinib [2], NSCLC tumors with *EGFR* exon 20 insertion (ex20ins) mutations are difficult to treat, demonstrating low response rates to those agents [3]. *EGFR* ex20ins mutations have been found in up to 12% of *EGFR*-mutated NSCLC tumors [4,5] and account for roughly 2% of all cases of NSCLC [5]. Most patients with *EGFR* ex20ins-positive (+) metastatic NSCLC (mNSCLC) receive first-line, platinum-based chemotherapy, but disease progression typically occurs within 6–7 months [3,6,7].

Mobocertinib is a first-in-class, oral EGFR TKI that selectively targets in-frame *EGFR* ex20ins mutations [8]. It was designed to form an irreversible covalent bond with cysteine 797 in *EGFR*, which leads to selectivity, high potency, and sustained EGFR kinase activity inhibition. In preclinical evaluation, mobocertinib showed better potency and selectivity for *EGFR* ex20ins mutations over wild-type *EGFR* compared with other TKIs, including erlotinib, gefitinib, afatinib, and osimertinib. Mobocertinib received accelerated approval from the US Food and Drug Administration (FDA) on 15 September 2021, for the treatment of adult patients with locally advanced or mNSCLC with *EGFR* ex20ins mutations whose disease has progressed on or after platinum-based chemotherapy [9]. Accelerated approval was based on the overall objective response rate (ORR) and duration of response (DoR) to mobocertinib 160 mg orally once daily demonstrated in an ongoing phase 1/2 study (NCT02716116). Durable responses were achieved in 114 patients with platinum-pretreated *EGFR* ex20ins+ mNSCLC, with an ORR of 28%, median DoR of 17.5 months, progression-free survival (PFS) of 7.3 months per independent review committee (IRC) assessments and an ORR of 35%, median DoR of 11.2 months, and PFS of 7.3 months per investigator assessments, and median overall survival (OS) of 24 months [10]. In 96 patients with previously treated *EGFR* ex20ins+ NSCLC in the EXCLAIM extension cohort of the same study, mobocertinib treatment resulted in a confirmed ORR of 25%, a median DoR not yet reached at the time of the analysis, and a median PFS of 7.3 months per IRC assessments, with median OS not yet reached, and a confirmed ORR of 32%, a median DoR of 11.2 months, and a median PFS of 7.3 months per investigator assessments [10].

The objective of the current analysis was to evaluate patient-reported outcomes (PROs) associated with mobocertinib 160 mg orally once daily in patients with previously treated *EGFR* ex20ins+ mNSCLC from the EXCLAIM extension cohort.

## 2. Materials and Methods

### 2.1. Study Design

The design of this three-part, open-label, phase 1/2 study containing dose-escalation/expansion cohorts and a single-arm extension cohort (EXCLAIM) was described in detail previously [10,11]. Briefly, the trial consisted of a dose-escalation phase, which included patients with advanced NSCLC who were refractory to standard therapies; an expansion phase, which included seven histologically and molecularly defined cohorts; and the EXCLAIM extension cohort, described here in more detail. In the current analysis of the EXCLAIM extension cohort, PROs of treatment with oral mobocertinib 160 mg once daily administered in 28-day cycles were assessed at 39 sites in Asia, Europe, and North America. Per protocol, PRO data were collected only in the extension phase. The study protocol and amendments were approved by the institutional review board or ethics committee at each study site, and study conduct complied with the principles set forth in the Declaration of Helsinki, International Council for Harmonisation Tripartite Guideline for Good Clinical Practice, and applicable local regulations. Written informed consent was obtained from all subjects prior to participation.

### 2.2. Participants

Eligibility criteria for the EXCLAIM extension cohort were described previously [10,11]. Briefly, eligible patients were ≥18 years of age and had NSCLC with a documented *EGFR* exon 20 in-frame insertion either alone or in combination with other *EGFR* or human epidermal growth factor receptor 2 (*HER2*) mutations and had received at least one prior line of therapy for locally advanced or metastatic disease. Patients with known active brain metastases, spinal cord compression, or leptomeningeal disease were excluded.

### 2.3. Patient-Reported Outcomes Assessments

PROs were used to evaluate key aspects of the patient experience while receiving treatment with mobocertinib. Secondary PRO measures included the European Organisation for Research and Treatment of Cancer (EORTC) Core Quality-of-Life Questionnaire (QLQ-C30) v3.0 and the 13-item EORTC lung cancer module (QLQ-LC13) v1.0. Dyspnea, cough, and chest pain were prespecified symptoms of interest. Exploratory PRO measures included the EuroQol-5 Dimensions-5 Levels (EQ-5D-5L) and Patient-Reported Outcomes Version of the Common Terminology Criteria for Adverse Events (PRO-CTCAE) questionnaire. HRQoL questionnaires were administered before any clinical measurements, assessments, or procedures were performed at the baseline visit, selected visits during the study, and 30 days after the last dose of mobocertinib. 

The EORTC QLQ-C30 v3.0 [12], which contains a global health status scale, five functional scales (Physical, Emotional, Role, Cognitive, Social), three symptom scales (Fatigue, Nausea and vomiting, Pain), and six single items (Dyspnea, Constipation, Diarrhea, Insomnia, Appetite loss, Financial difficulties), was used to assess cancer-related functions and symptoms. Patients used a 4-point Likert scale (“Not at all” to “Very much”) and a 7-point numeric rating scale (“Very poor” to “Excellent”) to rate their experiences during the past week. Higher scores on the global health status and functional scales and lower scores on the symptom scales correspond to better HRQoL.

The QLQ-LC13 v1.0 [13], a questionnaire constructed in parallel with the EORTC QLQ-C30, was used to assess lung cancer-specific symptoms (dyspnea, cough, hemoptysis, chest pain, arm or shoulder pain, pain in other parts), treatment-related side effects (sore mouth, dysphagia, peripheral neuropathy, alopecia), and use of pain medication. Patients rated their experiences during the previous week on a 4-point Likert scale (“Not at all” to “Very much”), with lower scores corresponding to better HRQoL. A composite score of dyspnea, cough, and chest pain from QLQ-LC13 was analyzed; improvement (or deterioration) was defined as the earliest of any of dyspnea, cough, and chest pain scores had improved (or deteriorated) by ≥10 points from baseline. 

General HRQoL was assessed with the EQ-5D-5L questionnaire [14], a preference-based measure of health status that has dimensions of Mobility, Self-care, Usual activities, Pain/discomfort, and Anxiety/depression, each containing 5 response levels (no problems, slight problems, moderate problems, severe problems, extreme problems) as well as a visual analog scale (VAS) to rate health from best (100) to worst (0). 

The PRO-CTCAE questionnaire, a patient-reported measurement system containing a library of 124 items representing 78 symptomatic toxicities, was designed as a companion to the National Cancer Institute (NCI)-CTCAE and developed to assess symptomatic toxicity in patients with cancer in clinical trials. In the EXCLAIM extension cohort, selected items (difficulty swallowing, decreased appetite, nausea, vomiting, diarrhea, rash, fatigue, and dry skin) from the PRO-CTCAE were administered to assess patient-reported adverse events. Symptom frequency, interference, and severity were evaluated.

### 2.4. Statistical Analysis

Patient-reported symptoms, functioning, and HRQoL on the EORTC QLQ-C30 and the EORTC QLQ-LC13 were secondary endpoints; patient-reported health status on the EQ-5D-5L and symptomatic treatment-related toxicity with selected items from the PRO-CTCAE were exploratory endpoints of the EXCLAIM extension cohort. PROs were analyzed in patients with baseline and at least one post-baseline measurement. The main PRO endpoints of interest were the core symptoms of lung cancer (e.g., dyspnea, cough, and chest pain), fatigue, physical functioning, and global health status/HRQoL. Actual values and changes from baseline on the EORTC QLQ-C30, the EORTC QLQ-LC13, and the EQ-5D-5L, and the selected items from the PRO-CTCAE were summarized with descriptive statistics over time. Least squares mean (LSM) changes from baseline and corresponding 95% confidence intervals (CIs) and *p* values on the EORTC QLQ-C30 and the EORTC QLQ-LC13 were obtained with linear mixed models that included baseline score and visit as covariates. The numbers and percentages of patients with improved and deteriorated EORTC QLQ-C30 and EORTC QLQ-LC13 subscale scores were also summarized over time. Clinically meaningful improvements were defined as a ≥10-point decrease from baseline, and deterioration was defined as a ≥10-point increase from baseline in symptom scores. Time to QoL improvement/deterioration was defined from first dosing date to the date of the first occurrence of QoL score improvement/deterioration of ≥10 points from baseline. For the PRO-CTCAE items, worsening to the two highest categories was summarized with descriptive statistics. The two highest response categories for level of severity were “severe” and “very severe,” for interference were “quite a bit” and “very much,” and for frequency were “frequently” and “almost constantly.” The percentage of compliance with the EORTC QLQ-C30, EORTC QLQ-LC13, EQ-5D-5L, and PRO-CTCAE was summarized over time. SAS version 9.4 or later (SAS Institute, Cary, NC, USA) was used for all statistical analyses.

## 3. Results

### 3.1. Participants

The EXCLAIM extension cohort included 96 patients with previously treated *EGFR* ex20ins+ locally advanced or mNSCLC; of these, 86 patients had been previously treated with platinum-based chemotherapy. At data cutoff (1 November 2020), 25/96 (26%) patients remained on mobocertinib treatment. The median time on treatment was 6.8 (range, 0.0–18.8) months. Median follow-up was 13.0 (range, 0.7–18.8) months. Across all PRO instruments, compliance rates were >90% at all time points, except at the end of treatment and 30 days after the last dose (Appendix A). Demographics and baseline characteristics of the patients in the EXCLAIM cohort are shown in Table 1. 

### 3.2. Effect of Mobocertinib Treatment on Core Symptoms of Lung Cancer

A total of 90 patients had a baseline and at least one post-baseline PRO measurement. LSM changes from baseline in the EORTC QLQ-LC13 subscale scores for dyspnea, cough, and chest pain are shown in Table 2, and overall mean changes from baseline in these scores over time are shown in Figure 1A. Mean improvements from baseline were evident at cycle 2 and maintained throughout treatment. More than 50% of patients had stable or improved symptoms of dyspnea, cough, and chest pain throughout the study cycles (Figure 1B–D). Clinically meaningful improvements were observed for dyspnea in 54.4% of patients, for cough in 46.7% of patients, and for chest pain in 38.9% of patients; 82.2% of patients had clinically meaningful improvement in the composite score, which comprised dyspnea, cough, and chest pain, with a median time to improvement of approximately 1 month. In patients in the EXCLAIM cohort who received platinum-based chemotherapy (*n* = 86), similar improvements in the core symptoms of lung cancer were observed on the EORTC QLQ-LC13.

### 3.3. Effect of Mobocertinib Treatment on Global Health Status/QoL and Other Functions and Symptoms

LSM changes from baseline in EORTC QLQ-C30 global health status/QoL during mobocertinib treatment are shown in Table 3. Baseline scores were maintained over time in most patients with mobocertinib therapy (LSM change from baseline, −1.8; 95% CI, −4.8, 1.2; *p* = 0.235). Approximately 41 patients experienced improvement in global health status, with a median time to improvement of 8.31 months. 

Significant worsening from baseline was observed for diarrhea (LSM change from baseline +34.1; 95% CI, 29.9, 38.2; *p* < 0.001) and appetite loss (+6.6; 95% CI, 2.2, 10.9; *p* = 0.004) (Table 3). Significant improvement from baseline was observed for dyspnea (−5.1; 95% CI, −8.1, −2.0; *p* = 0.002), insomnia (−6.5; 95% CI, −10.4, −2.7; *p* = 0.001), and constipation (−5.7; 95% CI, −7.9, −3.6; *p* < 0.001). LSM changes from baseline in the other subscale scores were not statistically significant. Thus, despite the worsening of some gastrointestinal symptom scores, particularly diarrhea, the global health status score and the summary score of all domains were maintained. The patients in the EXCLAIM cohort who received platinum-based chemotherapy showed similar maintenance/improvement of global health status/QoL on the EORTC QLQ-C30. 

### 3.4. Effect of Mobocertinib Treatment on General HRQoL

Patient-reported general HRQoL, as reflected by EQ-5D-5L VAS scores, remained stable during mobocertinib treatment (Figure 2). Among 90 patients with baseline data, the mean score was 74.28, and on day 1 of cycle 15 (last cycle with more than one patient completing the EQ-5D-5L VAS) among 17 patients, the mean score was 81.18, for an overall mean change from baseline of 5.88.

### 3.5. Patient-Reported Symptomatic Treatment-Related Toxicity

Worst symptomatic AEs for selected PRO-CTCAE symptoms through cycle 6, day 1 are shown in Figure 3. Most adverse events reported by patients were not in the two highest categories (“severe” and “very severe” for level of severity; “quite a bit” and “very much” for interference; “frequently” and “almost constantly” for frequency). At cycle 6, day 1, stable or improving symptoms were observed in most patients for all selected symptoms, except diarrhea and dry skin (Figure 4), with similar trends observed in cycle 2. Overall, 64.4% of patients experienced worsening of diarrhea frequency and 67.8% of patients experienced worsening of dry skin severity through the first 24 weeks of treatment. 

## 4. Discussion

The EXCLAIM extension cohort of this phase 1/2 study included 96 previously treated patients with *EGFR* ex20ins+ mNSCLC (86 patients had received prior platinum-based chemotherapy). In the current analysis of EXCLAIM (*n* = 90), PROs were used to evaluate key aspects of the patient experience while receiving once-daily treatment with mobocertinib, a first-in-class EGFR TKI that targets *EGFR* ex20ins mutations [8]. PROs were not evaluated in the platinum-pretreated patient cohort (N = 114) of the phase 1/2 study because some patients were enrolled in the dose-escalation and dose-expansion phases rather than EXCLAIM. 

Treatment with oral mobocertinib once daily for a median (range) of 6.8 (0.0–18.8) months resulted in clinically meaningful improvements in the core symptoms of lung cancer (dyspnea, cough, and chest pain) assessed by the EORTC QLQ-LC13, improved symptoms of dyspnea, insomnia, and constipation and maintained global health status as assessed by the EORTC QLQ-C30, and maintained general HRQoL assessed by the EQ-5D-5L VAS. Improvements in core lung cancer symptoms were observed within 2 months of treatment initiation and were maintained over time during treatment. These improvements were consistent with the clinical activity of mobocertinib observed in this difficult-to-treat population and suggest improvement of lung cancer-related symptoms. Mobocertinib therapy resulted in significantly worsening symptoms of diarrhea and appetite loss, as demonstrated on the EORTC QLQ-C30, as well as worsening symptoms of decreased appetite, diarrhea, dry skin, and rash reported by patients on the PRO-CTCAE. However, despite these worsening symptoms, global health status/QoL, physical functioning, emotional functioning, role functioning, cognitive functioning, and fatigue did not change significantly from baseline throughout mobocertinib therapy.

Data regarding HRQoL specifically in patients with *EGFR* ex20ins+ mNSCLC are limited. In a 2018 study interviewing 10 patients with mNSCLC (9 of whom were *EGFR* ex20ins+) [15], the most frequently reported symptoms included fatigue (90%), pain (70%), shortness of breath (70%), and cough (60%). Furthermore, all patients reported adverse psychological and emotional impacts of their disease, including anxiety about treatment, the future, or finances, as well as negative effects on activities of daily living. Mobocertinib, a novel targeted therapy, resulted in clinically meaningful improvement in core lung cancer symptoms and maintained general HRQoL in patients with *EGFR* ex20ins+ mNSCLC.

Limitations of this study include the single-arm, open-label design of the phase 1/2 study, including the EXCLAIM cohort, lack of a control group, and the small number of patients contributing data at the later time points. In addition, because this was a single-arm trial, HRQoL endpoints were prespecified but no formal hypothesis was made. Further study in a larger-scale, double-blind trial with an active comparator, as well as collection of HRQoL data in the real-world setting, would help to clarify the promising findings of this analysis. 

In conclusion, in this HRQoL analysis of previously treated patients with *EGFR* ex20ins+ mNSCLC, the first such analysis for an oral exon 20–targeted therapy, PRO results with mobocertinib treatment showed improvements in core lung cancer symptoms and maintenance of overall HRQoL and functions, despite AEs, such as diarrhea, dry skin, rash, and decreased appetite.

## Figures and Tables

**Figure 1 jcm-12-00112-f001:**
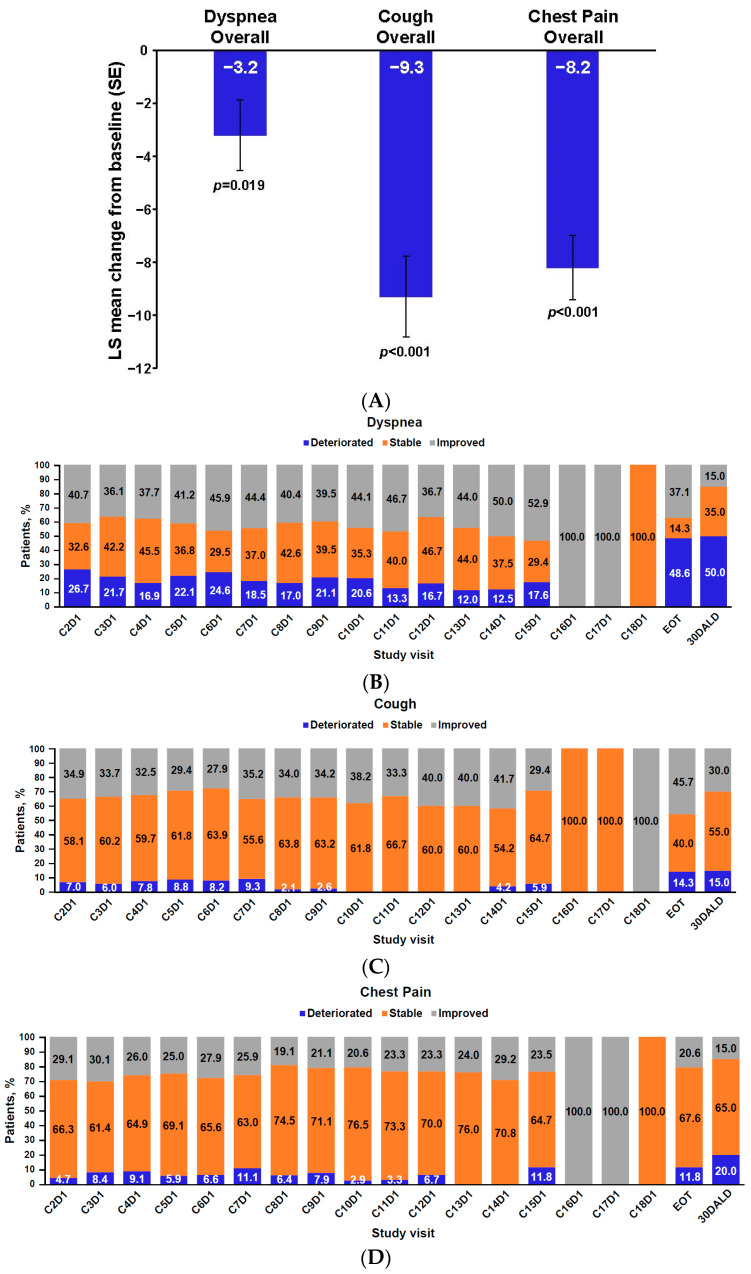
Change from baseline in EORTC QLQ-LC13 lung cancer symptom scores (**A**) and improvement or deterioration in EORTC QLQ-LC13 scores over time for dyspnea (**B**), cough (**C**), and chest pain (**D**) in the EXCLAIM cohort (N = 90). 30D ALD, indicates 30 days after last dose; C, cycle; D, day; EORTC QLQ-LC13, European Organisation for Research and Treatment of Cancer Core Quality-of-Life Questionnaire lung cancer module; EOT, end of treatment; LS, least squares; SE, standard error.

**Figure 2 jcm-12-00112-f002:**
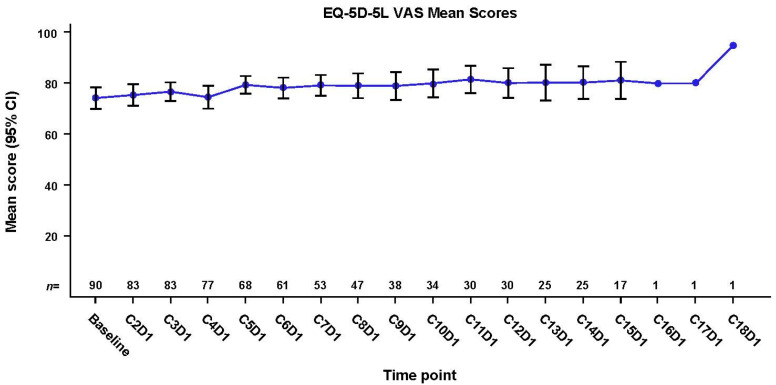
Mean EQ-5D-5L VAS scores over time. C, cycle; CI, confidence interval; D, day; EQ-5D-5L, EuroQol-5 Dimensions-5 Levels questionnaire; VAS, visual analog scale.

**Figure 3 jcm-12-00112-f003:**
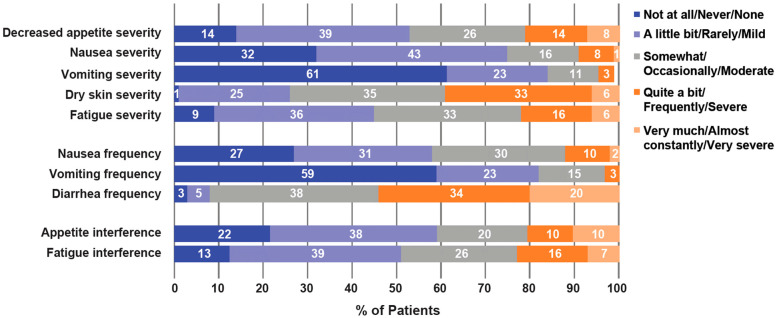
Worst reported symptomatic AEs on PRO-CTCAE through cycle 6, day 1 (*n* = 54); The PRO-CTCAE frequency of symptoms response options are never, rarely, occasionally, frequently, and almost constantly. For severity items: none, mild, moderate, severe, and very severe. For interference: not at all, a little bit, somewhat, quite a bit, and very much. One patient did not answer for vomiting severity. PRO-CTCAE, Patient-reported Outcomes Version of the Common Terminology Criteria for Adverse Events.

**Figure 4 jcm-12-00112-f004:**
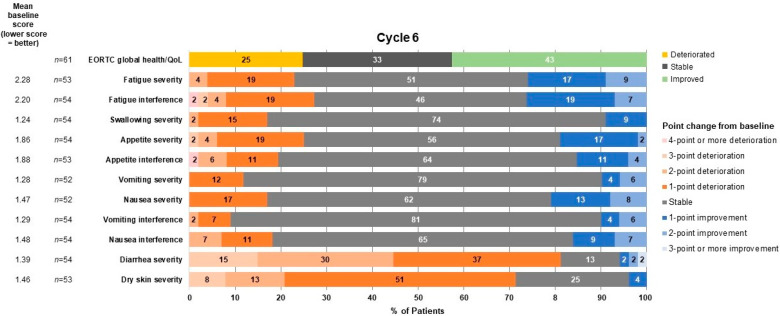
Percentage of patients with improvement, no change, or worsening in selected PRO-CTCAE symptoms at cycle 6, day 1. EORTC, European Organisation for Research and Treatment of Cancer; PRO-CTCAE, Patient-reported Outcomes Version of the Common Terminology Criteria for Adverse Events; QoL, quality of life.

**Table 1 jcm-12-00112-t001:** Demographics and Baseline Characteristics.

Characteristic	EXCLAIM Cohort (N = 96)
Median age (range), y	59 (27–80)
Female, *n* (%)	62 (65)
Race, *n* (%)	
Asian	66 (69)
White	28 (29)
Black	2 (2)
Other	0 (0)
Histology, *n* (%)	
Adenocarcinoma	95 (99)
Squamous carcinoma	1 (1)
Large cell carcinoma	0 (0)
ECOG PS, *n* (%)	
0	28 (29)
1	68 (71)
Smoking history, *n* (%)	
Never	70 (73)
Current	2 (2)
Former	24 (25)
Prior systemic anticancer regimens, *n* (%)	
1	49 (51)
2	30 (31)
≥3	17 (18)
Median number of prior regimens, *n*	1
Prior platinum-based chemotherapy, *n* (%)	86 (90)
Prior immunotherapy, *n* (%)	33 (34)
Prior EGFR TKI, *n* (%)	30 (31)
Baseline stable brain metastases, *n* (%)	33 (34)

ECOG PS, Eastern Cooperative Oncology Group performance status; EGFR, epidermal growth factor receptor; TKI, tyrosine kinase inhibitor.

**Table 2 jcm-12-00112-t002:** Change from Baseline in EORTC-QLQ-LC13 Lung Cancer Symptom Scores.

	LSM Change From Baseline [95% CI]		
Timepoint	Dyspnea Subscale	Cough Subscale	Chest Pain Subscale
C2D1	−4.4 [−7.6, −1.2] (*n* = 86)	−10.1 [−13.9, −6.3] (*n* = 86)	−10.9 [−14.3, −7.6] (*n* = 86)
C3D1	−5.2 [−8.4, −2.0] (*n* = 83)	−9.5 [−13.4, −5.7] (*n* = 83)	−9.0 [−12.4, −5.6] (*n* = 83)
C4D1	−3.8 [−7.1, −0.5] (*n* = 77)	−8.5 [−12.4, −4.6] (*n* = 77)	−8.6 [−12.1, −5.1] (*n* = 77)
C5D1	−4.9 [−8.3, −1.5] (*n* = 68)	−7.5 [−11.5, −3.4] (*n* = 68)	−8.2 [−11.8, −4.5] (*n* = 68)
C6D1	−3.9 [−7.4, −0.4] (*n* = 61)	−7.4 [−11.7, −3.2] (*n* = 61)	−7.7 [−11.5, −3.9] (*n* = 61)
C7D1	−4.3 [−7.9, −0.7] (*n* = 54)	−9.9 [−14.3, −5.5] (*n* = 54)	−6.5 [−10.5, −2.5] (*n* = 54)
C8D1	−2.9 [−6.7, 0.9] (*n* = 47)	−12.0 [−16.6, −7.4] (*n* = 47)	−6.1 [−10.3, −1.9] (*n* = 47)
C9D1	−1.9 [−5.9, 2.1] (*n* = 38)	−9.9 [−14.8, −4.9] (*n* = 38)	−8.0 [−12.6, −3.5] (*n* = 38)
C10D1	−3.5 [−7.7, 0.7] (*n* = 34)	−12.3 [−17.4, −7.1] (*n* = 34)	−10.3 [−15.0, −5.5] (*n* = 34)
C11D1	−3.7 [−8.1, 0.7] (*n* = 30)	−12.6 [−18.0, −7.2] (*n* = 30)	−9.3 [−14.3, −4.3] (*n* = 30)
C12D1	−1.2 [−5.6, 3.2] (*n* = 30)	−13.7 [−19.1, −8.3] (*n* = 30)	−6.4 [−11.5, −1.4] (*n* = 30)
C13D1	−3.5 [−8.2, 1.1] (*n* = 25)	−13.0 [−18.8, −7.2] (*n* = 25)	−8.3 [−13.7, −2.9] (*n* = 25)
C14D1	−3.4 [−8.2, 1.3] (*n* = 24)	−11.8 [−17.6, −5.9] (*n* = 24)	−11.1 [−16.6, −5.6] (*n* = 24)
C15D1	−3.7 (−9.0, 1.7) (*n* = 17)	−11.0 [−17.7, −4.3] (*n* = 17)	−8.0 [−14.3, −1.6] (*n* = 17)
C16D1	−7.5 (−27.1, 12.1) (*n* = 1)	−13.2 [−38.4, 11.9] (*n* = 1)	−9.8 [−34.5, 14.9] (*n* = 1)
C17D1	−7.5 (−27.1, 12.1) (*n* = 1)	−13.2 [−38.4, 11.9] (*n* = 1)	−9.8 [−34.5, 14.9] (*n* = 1)
C18D1	−3.9 (−23.5, 15.7) (*n* = 1)	−25.2 [−50.3, 0] (*n* = 1)	−8.8 [−33.6, 15.9] (*n* = 1)
EOT	3.3 (−0.9, 7.4) (*n* = 35)	−8.0 [−13.1, −2.9] (*n* = 35)	−6.3 [−11.1, −1.5] (*n* = 34)
30D ALD	4.9 (−0.2, 10.1) (*n* = 20)	−1.8 [−8.2, 4.6] (*n* = 20)	−1.0 [−7.0, 5.1] (*n* = 20)

30D ALD, indicates 30 days after last dose; C, cycle; CI, confidence interval; D, day; EORTC QLQ-LC13, European Organisation for Research and Treatment of Cancer Core Quality-of-Life Questionnaire lung cancer module; EOT, end of treatment; LSM, least squares mean.

**Table 3 jcm-12-00112-t003:** EORTC QLQ-C30 Overall Score Change from Baseline *.

Measure	N	LSM (SE)	95% CI	*p* Value
Global health status/QoL	86	−1.8 (1.50)	(−4.8, 1.2)	0.235
Physical functioning	86	0 (1.33)	(−2.7, 2.6)	0.986
Role functioning	86	−2.4 (1.76)	(−5.9, 1.1)	0.178
Emotional functioning	86	0.1 (1.26)	(−2.4, 2.6)	0.920
Cognitive functioning	86	−2.3 (1.29)	(−4.9, 0.2)	0.072
Social functioning	86	1.0 (1.80)	(−2.6, 4.6)	0.576
Fatigue	86	1.5 (1.6)	(−1.6, 4.7)	0.335
Nausea and vomiting	86	0.6 (1.12)	(−1.6, 2.9)	0.576
Pain	86	−3.3 (1.74)	(−6.8, 0.1)	0.060
Dyspnea	86	−5.1 (1.54)	(−8.1, −2.0)	0.002
Insomnia	86	−6.5 (1.93)	(−10.4, −2.7)	0.001
Appetite loss	86	6.6 (2.19)	(2.2, 10.9)	0.004
Constipation	86	−5.7 (1.08)	(−7.9, −3.6)	<0.001
Diarrhea	86	34.1 (2.10)	(29.9, 38.2)	<0.001
Financial difficulties	86	0.5 (1.85)	(−3.2, 4.2)	0.786

* Higher values for the global and functional domains indicate higher QoL or functioning; higher values on the symptom scales indicate higher levels of symptomology or problems. CI, confidence interval; EORTC QLQ-C30, European Organisation for Research and Treatment of Cancer Core Quality-of-Life Questionnaire; LSM, least-squares mean; QoL, quality of life; SE, standard error.

## Data Availability

The data sets, including the redacted study protocol, redacted statistical analysis plan, and individual participant data supporting the results of the completed study, will be made available after the publication of the final study results within three months from initial request, to researchers who provide a methodologically sound proposal. The data will be provided after de-identification, in compliance with applicable privacy laws, data protection, and requirements for consent and anonymization.

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
