# Peer review of "Mobocertinib (TAK-788) in EGFR Exon 20 Insertion+ Metastatic NSCLC: Patient-Reported Outcomes from EXCLAIM Extension Cohort"

_jcm, 2022, doi:10.3390/jcm12010112_

Round 1

Reviewer 1 Report (Previous Reviewer 1)

I thank the authors for providing a detailed revised version of Patient-Reported outcomes analyses from the EXCLAIM extension cohort. 

The authors provided, in this revised version, substantial clarifications, and I like a great deal the new figure 4.

Most clinicians do not understand QoL analyses easily, which is normal given it is very technical, and it takes a lot of time to learn and appraise. As such, care providers tend to rely on abstract or conclusion. Therefore, a clear presentation of the data is critical, specifically in abstract and conclusion sections. 

In this case, I still think the abstract would benefit from a clearer separation between results which are clinically meaningful, and those who are not. For instance, the fact that the only clinically meaningful difference to be detected in symptom scores was a detrimental effect in diarrhea is not enough clearly reported in the asbtract. The scientific community would greatly benefit from a better reporting of the abstract, and the trial and the results would be even more impactful.

Author Response

Reviewer 2 Report (Previous Reviewer 2)

Thank you for allowing me to re-review this paper. I believe that our suggestions have been adequately addressed and this paper is ready for publication. 

Author Response

Reviewer 3 Report (Previous Reviewer 3)

The authors have sufficiently addressed my comments. 

Author Response

This manuscript is a resubmission of an earlier submission. The following is a list of the peer review reports and author responses from that submission.

Round 1

Reviewer 1 Report

(see attached file)

Reviewer 2 Report

Thank you for allowing me to review the manuscript "Mobocertinib (TAK-788) in EGFR exon 20 insertion+ metastatic NSCLC: Patient-reported outcomes from EXCLAIM extension cohort" by Dr Campelo and colleagues.

Overall, the paper is well written. Tables and figures are excellent. I would suggest the authors to describe the limitations of the paper (phase 1/2 study with no control arm, QOL may be difficult to evaluate in the context of a clinical trial, etc) and the importance of collecting data in the real world setting as well. 

Reviewer 3 Report

The manuscript “Mobocertinib (TAK-788) in EGFR Exon 20 Insertion+ Metastatic 2 NSCLC: Patient-Reported Outcomes from EXCLAIM Extension 3 Cohort” by Garcia Campelo et al. reports patient-reported outcomes after treatment of patients with locally advanced or metastatic non-small cell lung cancer with EGFR exon 20 insertion (EGFRex20ins) with the first-in-class TKI mobocertinib.  In this study, mobocertinib was used as second line therapy and the majority of the patients (86 of 96) had been pre-treated with platinum-based therapy. Before every cycle of therapy, patients filled out the European Organisation for Research and Treatment of Cancer (EORTC) Quality of Life (QOL) Questionnaire-C30 (QLQ-C30), EORTC QOL Questionnaire, lung cancer module (QLQ-LC13), and General HRQoL assessed by the EuroQol-5 Dimensions-5 Levels (EQ-5D-5L) questionnaire. Authors report an improvement of the core lung cancer symptoms and the overall health-related QoL, although adverse events including diarrhea and rash occurred among most of patients during the treatment. This referee believes that the reported findings are relevant, as targeted treatments for patients with an EGFRex20ins mutation are limited. The following minor issues are to be addressed: 

1. Abstract: The abbreviation HRQoL is to be written out in full in the abstract, to align this with other abb, clarifying this also for readers unfamiliar with the term. 

2. Study design. A brief description of the original study design (published in JAMA Oncol ‘21) could help to understand the differences between the EXCLAIM vs other cohorts. As the manuscript is now put together, it is a bit unclear for the reader to get a full picture.

3. Number of patients. There are 96 patients in the EXCLAIM cohort, but in further analyses only 86 are included, seemingly because these were platinum pre-treated. What happened with those other 10? A clarification of why they were excluded and more detailed description of what the numbers are based on would be useful.

4. Compound. Mobocertinib is a relatively new compound with unique mechanism of action. It would be useful for readers of this manuscript unfamiliar with this molecular that a short description of its development, mechanisms of action and selectivity for ex20ins over WT EGFR are included in the introduction. 

5. Comparison to osimertinib. Authors ought to discuss how data of the current study compare to similar studies, and PROMs, done with the third-generation osimertinib inhibitors, as this compound that also has efficacy for the EGFRex20ins mutation. Or at the very least, the mobocertinib treatment findings ought to be discussed in the context of other first- or second-line treatments options for patients carrying this mutation.

6. Conclusion. Only the AEs diarrhea and rash are explicitly mentioned, although appetite loss and dry skin were also significantly worsening. It is recommended to reiterate the full details of the results to make the conclusion more comprehensive.